# Pashto Handwritten Invariant Character Trajectory Prediction Using a Customized Deep Learning Technique

**DOI:** 10.3390/s23136060

**Published:** 2023-06-30

**Authors:** Fazli Khaliq, Muhammad Shabir, Inayat Khan, Shafiq Ahmad, Muhammad Usman, Muhammad Zubair, Shamsul Huda

**Affiliations:** 1Department of Computer Science, Islamia College University Peshawar, Peshawar 25000, Pakistan; fazlikhaliq94@gmail.com (F.K.); zubair@icp.edu.pk (M.Z.); 2Department of Computer Science, University of Buner, Buner 19290, Pakistan; shabiradam86@gmail.com; 3Department of Computer Science, University of Engineering and Technology, Mardan 23200, Pakistan; usman@uetmardan.edu.pk; 4Industrial Engineering Department, College of Engineering, King Saud University, P.O. Box 800, Riyadh 11421, Saudi Arabia; ashafiq@ksu.edu.sa; 5School of Information Technology, Deakin University, Burwood, VIC 3128, Australia; shamsul.huda@deakin.edu.au

**Keywords:** enhance PHWD-V2 dataset, Pashto handwriting trajectories, customized CNN, prediction, and recognition

## Abstract

Before the 19th century, all communication and official records relied on handwritten documents, cherished as valuable artefacts by different ethnic groups. While significant efforts have been made to automate the transcription of major languages like English, French, Arabic, and Chinese, there has been less research on regional and minor languages, despite their importance from geographical and historical perspectives. This research focuses on detecting and recognizing Pashto handwritten characters and ligatures, which is essential for preserving this regional cursive language in Pakistan and its status as the national language of Afghanistan. Deep learning techniques were employed to detect and recognize Pashto characters and ligatures, utilizing a newly developed dataset specific to Pashto. A further enhancement was done on the dataset by implementing data augmentation, i.e., scaling and rotation on Pashto handwritten characters and ligatures, which gave us many variations of a single trajectory. Different morphological operations for minimizing gaps in the trajectories were also performed. The median filter was used for the removal of different noises. This dataset will be combined with the existing PHWD-V2 dataset. Various deep-learning techniques were evaluated, including VGG19, MobileNetV2, MobileNetV3, and a customized CNN. The customized CNN demonstrated the highest accuracy and minimal loss, achieving a training accuracy of 93.98%, validation accuracy of 92.08% and testing accuracy of 92.99%.

## 1. Introduction

The Pashto language possesses a cursive writing style encompassing various literary genres such as history, epics, poetry, stories, fiction, nonfiction, and more. Unfortunately, the presence of these invaluable literary works has been gradually diminishing in contemporary books. To preserve the rich heritage of Pashto literature, native speakers are now turning to artificial intelligence (AI) techniques. Detecting and recognizing handwritten Pashto characters and ligatures presents challenges due to their subtle variations and modifications.

### 1.1. Pashto Language

Pashto, spoken in Afghanistan and parts of Pakistan (particularly the Khyber Pakhtunkhwa and Baluchistan provinces), is the language of the Pashtun ethnic group [1]. Pashto speakers are also known as Pakhtuns or Pashtuns [2]. It is the official language of Afghanistan and the second most widely spoken language in Pakistan. Due to migration and cultural ties, Pashto is also spoken in neighboring regions of Iran and Tajikistan.

### 1.2. Pashto Handwritings

Pashto literature includes various written works, with the oldest known book being “PataKhazana”, an anthology of Pashto poetry dating back to the 8th century. Another important work is “Khairul Bayan”, a prose book by the legendary Pashtun figure Pir Rokhan or Bayezid Ansari [3]. Pashto literature covers various topics such as politics, religion, poetry, music, athletics, and education. However, the availability of modern technology and language services, including translation, text recognition, and speech recognition, is limited for Pashto speakers [2].

### 1.3. Pashto Handwritten Characters and Ligatures Recognition

Recognizing handwritten Pashto characters and ligatures is a complex task in the era of artificial intelligence. Pashto has a unique set of 44 characters [4,5], including other alphabets introduced by Pir Rokhan [4]. These characters combine to form different words, and their positioning in a word generates various ligature shapes [2]. Compared to other scripts like Arabic, Urdu, and Persian, Pashto exhibits a more significant variation in character shapes and a higher number of ligature joints.

Despite the significance of Pashto, there is a lack of available Pashto handwritten datasets and limited research on Pashto OCR (optical character recognition) for handwritten character recognition. The recognition of Pashto characters and ligatures poses challenges due to the language’s diverse alphabet sets and the significant variation in character shapes based on word positions [6]. The partial view of Pashto handwritten ligatures is shown in Figure 1.

Every individual wants to contribute and do something new for their homeland, i.e., for their people, language, culture, education, and much more. It is a morally spiritual motivation to spend energy on the Pashto language. In this new era of artificial intelligence, there is a need for such research or study to facilitate Pashto speakers in learning and communication. This research is also helpful in translating Pashto into different languages based on recognition patterns.

The main contributions of this research are given in the following:▪A novel deep-learning-based model is proposed, which is lightweight and efficiently classifies and recognizes variational Pashto handwritten characters and the different shapes of characters concerning connectivity with each other. The Pashto character may have two to four possible shapes to construct a complete word, i.e., isolated, middle, first, and end.▪The second contribution is the construction of the Pashto handwritten character and ligature data set. The Pashto language is a low-resource language, and this paper also contributes to its resource generation. This dataset is different from Pashto character datasets because it also consists of the different shapes of a character.▪The shapes of Pashto characters and ligatures have been classified and recognized with geometric variation, i.e., rotation, location shifting, and scaling.

The proposed technique will motivate other researchers to generate different regional language resources and lightweight deep-learning techniques for their respective regional languages. Handwritten Pashto text classification and recognition have serious challenges, which are given as follows:Invalid hooks exist, which affect the accuracy of previously published techniques.Salt and pepper noise is generated during scanning and the type of written material.Zig-zag motion is generated due to hand shivering and writing speed that changes the shape and features of a language base symbol.Invalid disconnected strokes.Rotated characters and ligature.Variant size of the same character and ligature.The difference in the handwriting of the same characters and ligature.

This paper is further divided into sections as follows:

Section 2 discusses related work. Section 3 elaborates the proposed approach. Section 4 elaborates on the experimental results and discussion, whereas Section 5 is the conclusion.

## 2. Related Work

The Pashto language has many features and variations to be investigated, and many researchers have proposed different techniques to recognize Pashto text. Pashto text has been discussed elaborately in the paragraphs given below.

Recognizing handwritten letters, words, signatures, and maps is a significant and challenging issue. It has beneficial uses in various industries, including banking, retail, education, data gathering, and touchscreen gadgets [7]. Many researchers worked on creating and collecting Pashto Characters and their ligatures datasets of printed text. An immense amount of Pashto text is presented on the internet to acquire corpora for extracting valuable information to create different types of Pashto printed text datasets [6,8,9,10]. However, in [8], the authors created a dataset of whole words in printed form and then manually marked the segmentation points. They developed a corpus of 2,313,736 Pashto words from web sources. A total of 19,268 distinct ligatures are found, of which 7000 ligatures cover 91% of words.

A bidirectional and multi-dimensional long short-term memory (BLSTM and MDLSTM) network that recognizes printed Pashto text was made in [11]. The limitation of this work was its high constraint on location, i.e., baseline. In [6], the authors claim to construct a printed Pashto words ligature-based dataset having 8k images of 1k different ligatures. They performed manual augmentation, i.e., they showed different sizes, orientations, and positions. The authors in [9] proposed a holistic approach for recognizing printed Pashto words with 1k ligatures. The study also reveals that the SIFT descriptor performs better than standard feature extraction approaches, i.e., PCA.

Furthermore, different researchers created a Pashto single handwritten character dataset [1,2,12,13] with different implementation techniques and obtained different results. Researchers in [14] implemented HOG, which is helpful in various sorts of rotation because it is rotationally invariant. For 730 characters, the suggested approach obtained a maximum prediction accuracy of 93.5%. In contrast, in [5], the researcher generated a handwritten Pashto character dataset only having 11352 images and implemented different models, i.e., histogram-oriented gradients (HOG), having a calculated accuracy of 80.34%. In paper [13], zoning-based features have an estimated 76.42% accuracy using 10-fold cross-validation and, a low-level feature technique based on K-nearest neighbors has an accuracy of 74.8%. Another researcher in [15] implemented HoGs, the Gabor filter, DCT, DWT, and hybrid feature maps based on a zoning technique. Accuracy ratings of 63.30%, 65.13%, 68.55%, 68.28%, 67.02%, and 83% were achieved with different configurations. The final accuracy produced using convolutional neural networks was 81.02% lower than that of the multi-class support vector machine. The hybrid feature map-based multi-class SVM model had an accuracy of 83%. In [16], the trials’ continuous findings indicate that the recommended OCR system is 80.7% accurate, artificial neural networks are 78% accurate, and SVM is 56% accurate.

Another approach in [4], the probability-based multi-class naive Bayesian classifier, which computes the probabilities of geometric invariant properties to predict the highest likelihood, was used for real-time character recognition with a real-time accuracy of 97.5%. In paper [17], a dataset of 43,000 images with multiple rectified linear unit (ReLU) layers was trained and evaluated using three different feed-forward neural network configurations. Models using the backpropagation method were Model 1, using a single ReLU layer; Model 2, using two ReLU layers; and Model 3, incorporating three ReLU layers. According to the simulation, in contrast to Model 1’s accuracy of 87.6% on anonymous data, Model 2 and Model 3’s accuracies were 81.60% and 3%, respectively. In paper [1], recognizing Pashto handwritten characters suggested a convolutional neural network (CNN) model. The experimental results show that the suggested model was better than other models, with a test accuracy for PHCR (Pashto handwritten character recognition) of 99.64%, limited to only one character.

In the same way, in [18], a convolutional neural network was applied to detect offline Urdu handwritten letters in many fonts in an unrestricted setting. Un [19], multi-level sorting of the clustered data was one of the optimization steps used by the evolutionary algorithm to improve the classification criteria for identifying Urdu ligatures. Using the benchmark UPTI dataset, experiments produced a recognition rate of 96.72%. The summarize comparative analysis of the literature review is shown in Table 1. 

## 3. The Proposed Approach

For the conservation and preservation of Pashto handwritten text, a lightweight deep-learning technique has been designed to detect and recognize geometrically invariant handwritten characters and their positional ligatures. The Pashto language consists of 44 characters [20], from which all Pashto words are formed. Not every word has an objective meaning in Pashto, but the meanings of words change with different dialects and accents [11]. Each word forms a Pashto word according to a specific dialect or accent. These words are combined to form sentences like in other cursive script languages, i.e., Urdu, Arabic, Persian, etc. The terms “Pashto” or “Pakhtu” may also be known as Afghani throughout Indo-European regions and are denoted in phonetic international as ***p’**æ**khtu***. The various Indo-Iranian language families include the East Iranian branch which includes the Pashto languages. There are two main dialects of the Pashto language: one is called the soft dialect, and the other is called the hard dialect. The hard dialect is known as the “Northern dialect” and the softer dialect as the “Southern dialect”. These two dialects are phonetically different from each other. The word Pashto is pronounced “Pakhtu” in northern dialects but “Pashto” in southern dialects. Both hard and soft Pashto dialects are considered in this study. Another dialect that is considered the standard of the Pashto language is Kandahari Pashto [15]. Pashto words are challenging to recognize using optical character recognition (OCR). Researchers provided different techniques and scenarios to detect, classify or recognize Pashto handwritten characters and words. They form a significant development and will be a major leap forward for the future success of the Pashto language.

### 3.1. Pashto Handwritten Ligatures

Pashto’s characters have a specific writing shape while constructing the words. The change in a character’s shape is due to the cursive property of Pashto handwriting. Ligatures mean that the character’s different variations are connected at different locations. In the Pashto language, the character can change meaning in different positions, i.e., a character at the start of a word has a different shape from when it is used as a single character. When a character is at the center or the end of a word, it generates a different style and pattern. All these patterns are the properties of Pashto and other cursive languages, as shown in Figure 2 below.

### 3.2. Dataset

As discussed earlier, a slight change in the structure of the Pashto characters makes classification and recognition more complex. A real Pashto handwritten character and ligature dataset is required as no such dataset exists up to the best of our knowledge. There are 44 characters and 110 ligatures. The ligatures are generated due to different positions in word construction. Some characters cannot connect with others in word construction and stand isolated. Therefore, the Pashto language generates only 110 unique ligatures, i.e., 44 characters, and 110 ligatures concerning character position in the word.

#### 3.2.1. Templates Designed for Data Collection

A template for the collection of handwritten Pashto characters and ligatures was made, which would eventually lead to the creation of a dataset. A sheet was created on A4-sized paper with a 10 × 10 grid, i.e., the grid consists of 10 rows and 10 columns. This grid would collect 100 different samples written by 100 different people. Still, for better results and training in deep learning techniques, the number of handwritten characters and ligatures was increased from 100 to 200 individuals, which included high school teachers and students of Grades 9 and 10. These individuals wrote a single Pashto character and ligature in the center of each cell. In the same way, 200 individual writing samples were generated, which produced 30,800 images.

#### 3.2.2. Generation and Collection of Datasets

After collection, the A4-sized paper with a 10 × 10 grid was then scanned on an HP Laserjet Pro MFP M426fdn printer. All these scans were placed in a folder, then all the images were cropped, and then the grid was removed with Photoshop, and only the characters and ligatures remained; the noise was also removed after the grid’s removal through deep-learning techniques, which are discussed in detail in the preprocessing section. After removing gridlines and noise, 10 × 10 images were created for each character. The image’s background changed from white to black, and the characters were changed from black to white for the enhancement of accuracy. Two-hundred images were created for each character and ligature, and each character was placed in a separate folder. The dataset consisted of 154 unique characters and ligatures, and 154 folders were created. Each folder contained two-hundred images of a single Pashto character and ligature. The dataset consisted of 30,800 images and 154 classes.

### 3.3. Preprocessing

Upon the collection and generation of the dataset, the data were put into the second phase of the deep-learning technique known as preprocessing. This step was performed to prepare the data in a pure form for implementation. Some of the problems that emerged are as follows.

#### 3.3.1. Missing Trajectories

This technique identified missing values in the trajectories, i.e., when a Pashto character or word is written, sometimes the connection point is missed, but the character and its meaning remain unchanged. According to the rules, the missing space should be where the system cannot recognize it. This problem is shown in Figure 3 below for a quick understanding.

For this, a deep-learning approach called image inpainting is better at replicating filled regions with fine details. The image inpainting technique, called Edge Connect, is a two-stage adversarial model that starts with an edge generator and ends with an image completion network [6,21]. The edge generators fill the gap and connect the characters based on its data set, and the image completion network prioritizes filling in the incomplete regions using visual hallucination edges [21]. The datasets used in the Edge Connect approaches are publicly available in CelebA, Places2, and Paris StreetView.

#### 3.3.2. Removal of Noise

Different noises were removed and cleaned from the dataset in the preprocessing step. For instance, one of the most popular order–statistic filters is the median filter because it handles certain forms of noise, including Gaussian, random, and salt and pepper noise. The median filter substitutes the median value of the relevant window for the center pixel of an M-by-M neighborhood. Be aware that noise pixels are thought to deviate significantly from the median. This kind of noise issue may be eliminated using the median filter technique. Before performing the binarization procedure, this filter is employed to eliminate the noise pixels from the protein crystal pictures [22]. Crystallization and crystal identification are essential phases in the experiment for increasing the accuracy of the picture classification and recognition algorithms.

#### 3.3.3. Minimization of Invalid Hooks

Unwanted signals, which are missed or exceed the base symbol of the Pashto character, are discussed. These structures are also generated in the dataset and trained with different models and techniques. Invalid hooks are generated during rushed Pashto handwriting. The Pashto language consists of different types of hooks, some of which are shown in Figure 4.

### 3.4. Customized Deep Learning-Based Techniques

After data cleaning and preprocessing, a customized lightweight deep-learning convolutional neural network was designed. The proposed technique was trained on the developed dataset, elaborated as the “neurons” that make up a neural network, i.e., a collection of interconnected nodes. The input, hidden, and output layers are where neurons are organized. The input layer represents the predictors/features, and the output layer represents the response variable(s). Convolutional operation involves multiplying arrays element by element and grouping or summing the result to produce a new array representing *a* × *b*, multiplying the components of matrix b by the first three elements of matrix *a* [16]. In the general architecture of CNN-based techniques, the convolution operation is calculated using (1).
(1)$$x_{ij}^{l}=\sum_{a=0}^{m-1}\sum_{b=0}^{m-1}\omega_{ab}y_{\left(i+a)(j+b\right)}^{l-1}$$

The product is added and saved in a new *a* × *b* array, demonstrating convolutional layers without interruption until the function is completed.

Convolution and pooling extract properties from data chains, with a multi-layer perceptron flattening the data. Max-pooling layers produce the maximum number in a region, without learning independently for some k×k region. For instance, if their input layer is an *N* × *N* layer, the output is generated as an Nk×Nk layer since the max function reduces each k×k block to a single value.

The activation function adds non-linearity to the network, using differentiable sigmoid functions and hyperbolic tangents. The output is transferred to the next layer, where ReLU is the most widely used feature. These functions generate outputs ranging from 0 to 1, with ReLU being the most widely used feature in deep learning. [18].

The SoftMax function, utilized for multi-class classification issues, was chosen for this project. It is a sigmoid function generalization. Furthermore, it generates outputs between 0 and 1.

The neuron displaying the input $\left(x_{1}-x_{2}\right)$, their associated weights $(w_{1}-w_{2}),$ a bias (shown in the image), and an activation function (shown in the figure) applied to the weighted sum of the inputs is given as follows:(2)$$f(b+\sum_{i=1}^{n}x\omega_{i})$$

Thus, five layers of CNN architecture are created, as shown in Figure 5 below.

After the data preprocessing step, the dataset is fed into the customized CNN architecture. The proposed customized CNN architecture has five convolution and five max-pooling layers, i.e., **Conv**→**Maxpooling**→**Conv**→**Maxpooling**→**Conv**→**Maxpooling**→**Conv**→**Maxpooling**→**Conv**→**Maxpooling**. After the last maxpooling layer, the resultant vectors are converted into a one-dimension feature vector. The feature vector is processed via an artificial neural network. The last layer gives the desired output of the invariant Pashto handwritten characters and ligatures, as shown in Figure 5.

### 3.5. Deep Learning Techniques Experimentation

After collecting the required datasets, the process of implementing different deep-learning techniques, as shown in Figure 6, obtained different results for detecting and recognizing Pashto handwritten characters and ligatures. Different algorithms were implemented, which produced different results on different epochs. They are discussed in detail below.

## 4. Experimental Results and Discussion

This study has used different deep-learning techniques, and each technique has presented different results. All the results were different, and each technique had a separate algorithm. All the deep-learning algorithms showed significant results, but the custom-made CNN model had even better results. The more we tried and experimented, the better the results we had [23].

### 4.1. Dataset Development Process

Many researchers have tried to resolve issues regarding character and ligature recognition using artificial intelligence, especially in the case of words written by hand. A Pashto character database was created by Uddin I. et al. [17] for the experimental and simulation work, which worked on the creation of OCR for handwritten Pashto characters. With the automated recognition of Pashto handwritten characters, Mudaser et al. [12] created a small-size database of Pashto handwritten characters. They used the following techniques: a recurrence neural network, a convolutional neural network, and a deep neural network.

According to the literature, finding a workable solution remains a significant challenge for researchers. Without a dataset of handwritten characters, developing an efficient solution for the identification and recognition of inconsistent handwritten Pashto text is quite difficult. This article describes the creation of a database to handle the issue of interpreting Pashto handwritten letters and ligatures. The three steps of the recommended database development process are data collecting, scanning and character extraction, and labelling and arrangement.

#### 4.1.1. Template Design Phase for Dataset Collection

For Pashto handwritten character and ligature detection and recognition, a data collection template was required to collect the required dataset. For this, a template was designed for the data collection of Pashto handwritten character and ligature samples on A4-sized paper. This template consisted of a 10 × 10 grid in which 200 samples of Pashto characters and ligatures were written.

#### 4.1.2. Template Printing and Required Classes

A template was printed for the dataset comprising 154 classes, consisting of Pashto handwritten characters and ligatures used in Pashto words. Hence, 308 sheets were printed.

#### 4.1.3. Data Collection

After printing the required template, a high school was visited where high school teachers and students were asked to participate in creating different character and ligature datasets. Almost 200 teachers and students contributed to creating the dataset, and the required dataset was generated in about one month. Based on their position, gender, and age, students’ and faculty members’ contributions are shown in Table 2.

The dataset was collected from different students and teachers and scanned through an HP Scanner, as shown in Figure 7.

#### 4.1.4. Gridline Removal Phase

After dataset collection and scanning, template grid removal was required to crop all the images. An app for removing salt and pepper noise already exists, but a system was needed to remove the gridlines from the sheet; consequently, photoshop was used as there are no applications for removing gridlines other than photo-editing apps. The following result is shown in Figure 8.

#### 4.1.5. Template Segmentation

The images were cropped according to the grid to achieve a single character and ligature dataset. Then, Python was used to develop and design a system that cropped the images in 10 × 10, i.e., one-hundred equal parts, and stored these 154 different classes into separate folders. Each folder consisted of one-hundred different images as a separate entity in JPG format, as shown in Figure 9.

#### 4.1.6. Morphological Operation

After cropping, the dataset obtained had much noise, like salt and pepper noise, because of the scanning process. The noise could be removed by applying different morphological and noise-removal techniques, such as dilation. The mathematical equation for dilation is
(3)$$A⊕B=U_{b\in B}A_{b}$$

The locus of the points covered by the center of B that move within A may be viewed as the dilatation of A by B, if B, as before, has a center at the origin [24]. f⊕s is the dilation of image f caused by structuring element s, whereas the mathematical formula for erosion is
(4)$$A⊖B=\{z\in E|B_{=}\subseteq A\}$$

If any pixel in a binary picture is set to 0, the output pixel is also set to 0. f⊖s represents the erosion of picture f caused by structuring element s. The new pixel value is established once the structural element s is positioned with its origin at (x, y) [24]. An image can become tainted by salt and pepper noise by having some pixel values randomly changed to 255 or 0. The interpolation method or filter design is the foundation of the conventional image-denoising algorithm. The convolutional neural network (CNN) is not eliminating salt and pepper noise itself [25]. The partial view of the prepared Pashto character and ligature dataset is shown in Figure 10.

### 4.2. Deep Learning Techniques Experimentation

After collecting the required datasets, different deep-learning techniques were implemented to obtain different results for detecting and recognizing Pashto handwritten characters and ligatures. Different algorithms were implemented, which produced different results on different epochs. They are discussed in detail below.

#### 4.2.1. VGG 19

Simonyan and Zisserman (2014) introduced VGG19, a convolutional neural network of 19 layers. Among the 19 layers, 16 are convolution layers, and three are fully connected layers that classify the images into 1000-item categories. The VGG19 algorithm is trained using the ImageNet database, which contains one million images organized into 1000 categories. Since each convolutional layer uses 33 filters, it is a well-liked method for classifying images [26]. The outcome of the VGG19 after 50 iterations was 0.1567 training loss, 0.9467 training accuracy, 0.7993 validation loss, and 0.8085 validation accuracy. This architecture did not produce better results because VGG19 requires extensive information resources through which this model can learn maximally. On the other hand, Pashto characters and ligatures have very little details, such as slight arches; as a result, VGG19 will produce good results in this scenario. The graph is shown in Figure 11a,b:

#### 4.2.2. MobileNetV2

MobileNetV2, a mobile architecture that improves the performance of mobile models for various sizes, operations, and benchmarks. There are two different kinds of blocks in MobileNetV2, i.e., those with a stride of 1, a residual block, and another block for shrinking with a stride of two. For both kinds of blocks, there are three levels. This time, 11 convolutions with ReLU6 make up the first layer. The depth-wise convolution is the second layer, and the additional one-to-one convolution is the third linear layer [27]. The result of MobileNetV2 after 50 epochs was a validation loss of 0.6854 and a validation accuracy of 0.8024, with a training loss of 0.2327 and a training accuracy of 0.9328. This technique does not converge in the validation phase efficiently because of the manifold of interest (MoI). The Pashto characters and their variant ligatures have small trajectories that contain fewer data, and this technique further reduces the number of features due to MoI [27]. The graphs for validation and training are shown in Figure 12a,b, respectively.

#### 4.2.3. MobileNetV3-Large

MobileNetV3 has two versions, MobileNetV3-Large and MobileNetV3-Small. Both approaches are intended for use in situations with plenty or little resources. The models are generated using Platform-aware NAS, NetAdapt, and the network improvements discussed in this section are also included [28]. MobileNetV3Large is used in both the training and validation phases using our dataset. MobileNetV3Large achieved a training accuracy of 0.9062 and a training loss of 0.2799. The same technique achieved a validation loss of 0.9823 and a validation accuracy of 0.7576 on development data. Another intriguing aspect of MobileNetV3 is that it has few parameters and depth, even though the number of parameters is substantially less and produces some noise [29]. Figure 13a displays the training and validation accuracy, while Figure 13b shows the training and validation loss.

#### 4.2.4. Proposed Technique

In the convolutional layer, we tuned the parameters with a stride of 1 because Pashto handwritten characters and ligature trajectories have very little information for detection and recognition. To increase accuracy, we extracted the fine details. The proposed technique used the same padding value, which lost some vital information, but in the case of Pashto handwritten trajectories, we used data augmentation as well and some trivial information was present in the boundary of the image. Kernel size was restricted to 3 × 3 because the size 5 × 5, 7 × 7 and 9 × 9 lost some information on stroke trajectory. The next customized entity for Pashto handwritten character and ligature trajectories was the number of filters for each layer. In the first layer, the number of filters was 32; for the second layer, the number of filters was 64; for the third layer, the number of filters was 128; for the fourth layer, the number of filters was 256; for the fifth layer, the number of filters was 512. In total, there were 992 different filters. The dropout value was fixed to 0.01 because we did not want to drop large numbers of neurons, as the number of classes is large, i.e., 154. In augmentation, eight parameters were applied, i.e., scaling, rotation, etc., and the learning rate used was 0.01. In the pooling operation, the proposed technique used the max-pooling operation. The hidden layers perform feature extraction by carrying out different computations. Here, the first layer consists of convolution, then a max-pooling layer, then, again, a convolution layer followed by a max-pooling layer, and then a ReLU. These five multiple hidden layers extract features from an image according to the aforementioned parameters. The result produced by the customized CNN architecture generated the best result. It outperformed different algorithms, as discussed above, with their results on Pashto handwritten characters and their ligature detection and recognition. The main reason for this enhanced result is that this architecture is designed explicitly for Pashto characters and ligatures. According to the given information, it produced state-of-the-art results. The result after 50 epochs produced a training accuracy of 0.9398, training loss of 0.1783, validation loss of 0.2573, and validation accuracy of 0.9208, as shown in Figure 14a,b.

A comparison of Pashto handwritten characters and their ligatures on different techniques is shown in Table 3. The customized CNN result is better than all other techniques.

## 5. Conclusions

Pashto handwritten character and ligature classification and recognition is a precious, challenging, and complex task. Different existing deep-learning and customized architectures were used to classify and recognize Pashto handwritten invariant characters and ligatures. The customized CNN model outperformed the existing deep-learning architectures in terms of accuracy and loss. The training and validation accuracies of the proposed customized CNN are 93.98% and 92.08%, respectively. Similarly, the training loss is 17.83%, and the validation loss is 25.73%. A trajectory-based dataset of the characters and their variants is developed, which marks a new contribution to the Pashto language’s resource generation. This research gives a new dimension to the Pashto handwritten text classification and recognition. Future work: The accuracy can be increased by adding more geometrically varied characters and ligatures. As people have diverse writing styles, adding diverse writing styles can improve the impact of the proposed technique.

## Figures and Tables

**Figure 1 sensors-23-06060-f001:**
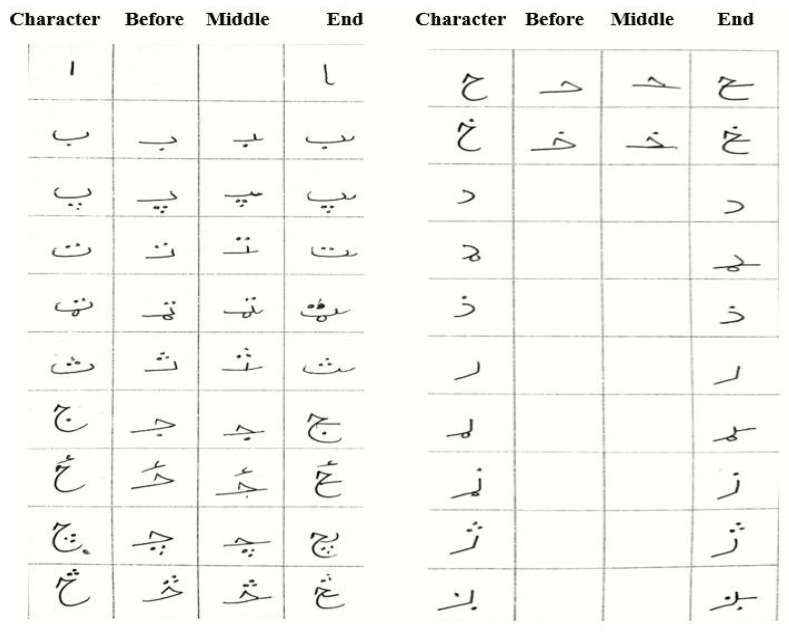
Partial view of Pashto handwritten ligatures.

**Figure 2 sensors-23-06060-f002:**
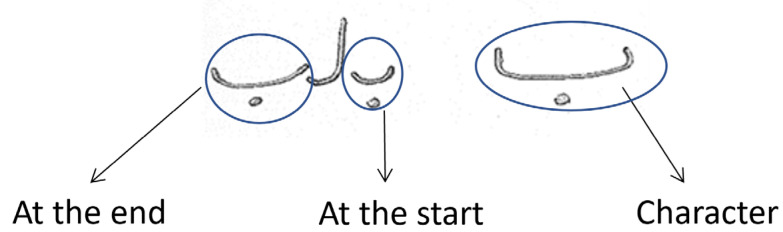
Pashto handwritten word ligatures.

**Figure 3 sensors-23-06060-f003:**
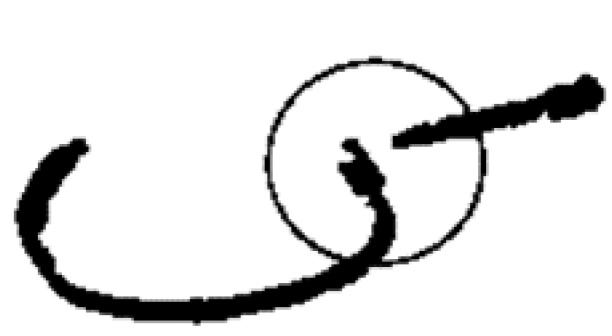
Pashto handwritten character dysconnectivity.

**Figure 4 sensors-23-06060-f004:**
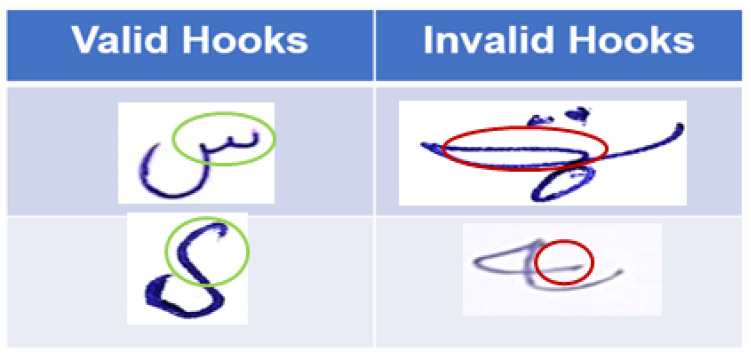
Valid and invalid hook detection and recognition.

**Figure 5 sensors-23-06060-f005:**
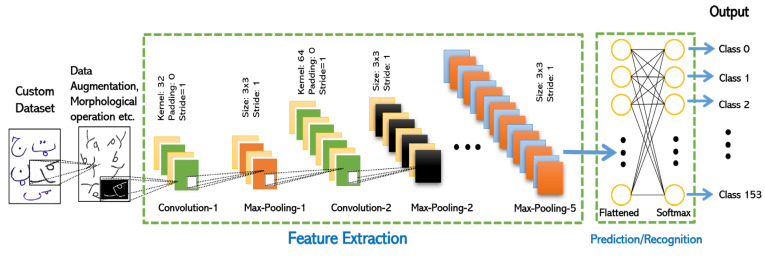
Customized 5-layer CNN model.

**Figure 6 sensors-23-06060-f006:**
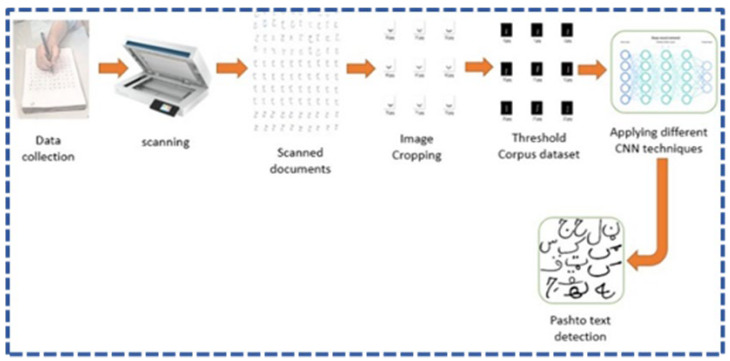
General overview of the proposed framework.

**Figure 7 sensors-23-06060-f007:**
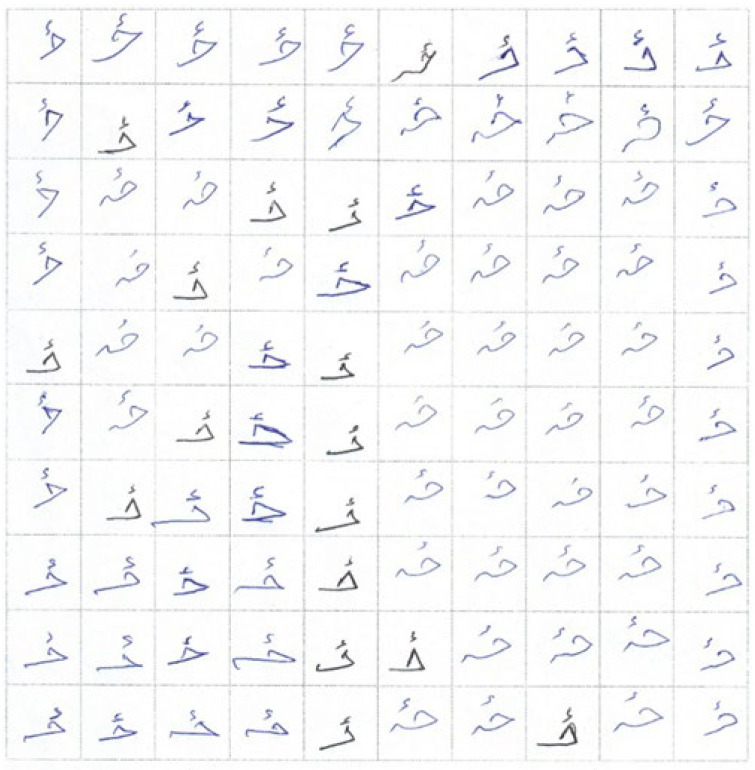
Dataset collection phase.

**Figure 8 sensors-23-06060-f008:**
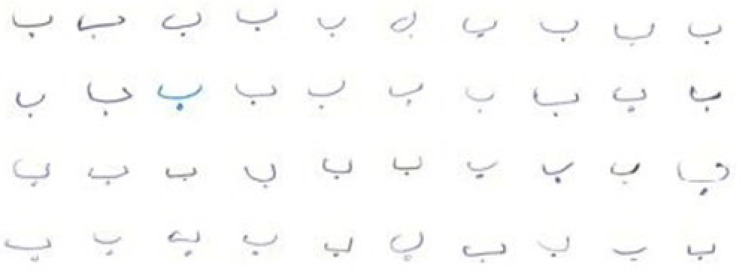
Partial view of dataset without gridlines.

**Figure 9 sensors-23-06060-f009:**
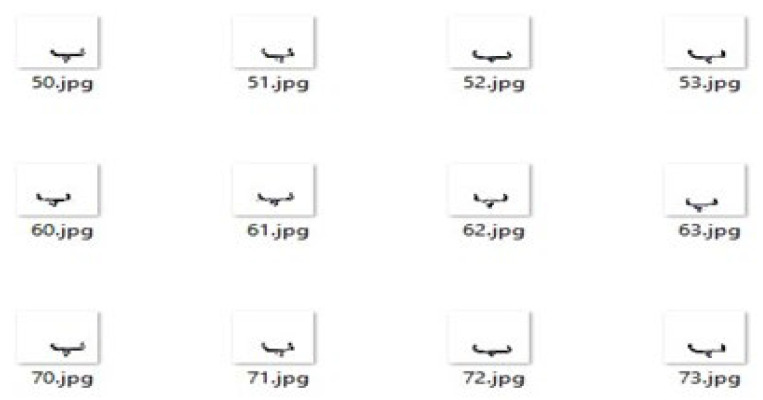
Partial view of dataset cropping phase.

**Figure 10 sensors-23-06060-f010:**
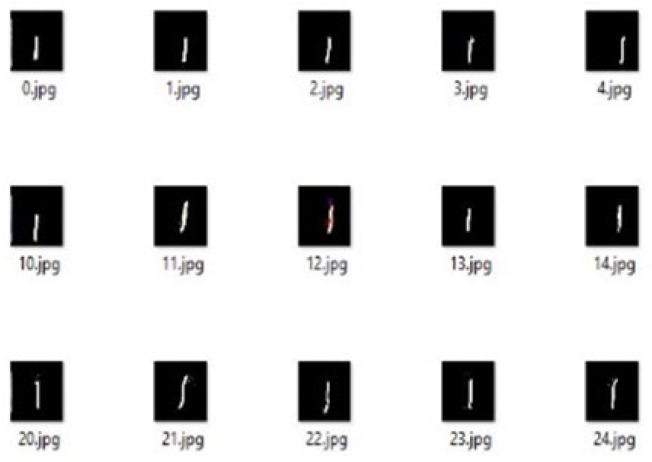
Partial view of noise-free image dataset.

**Figure 11 sensors-23-06060-f011:**
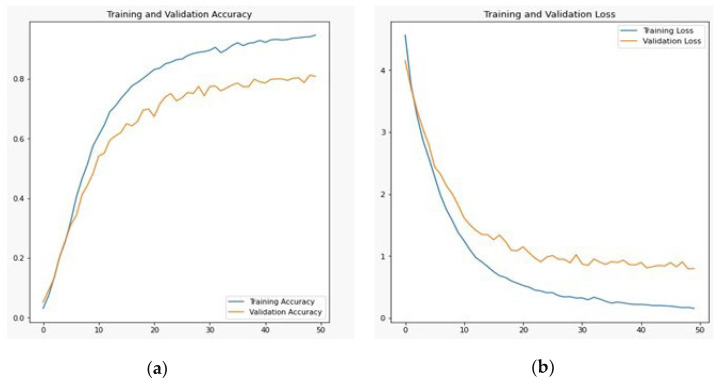
VGG19 training and validation (**a**) accuracy and (**b**) loss.

**Figure 12 sensors-23-06060-f012:**
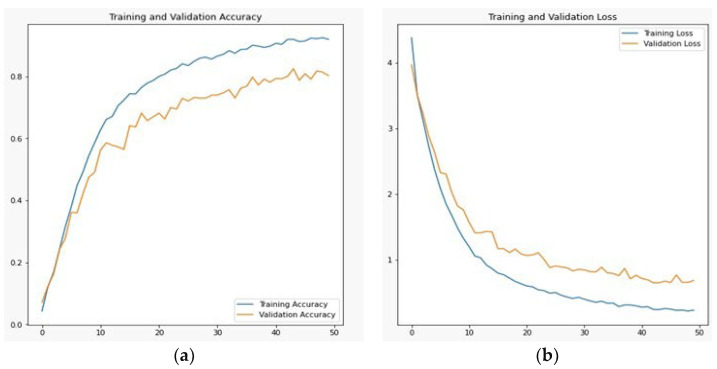
MobileNetV2 training and validation (**a**) accuracy and (**b**) loss.

**Figure 13 sensors-23-06060-f013:**
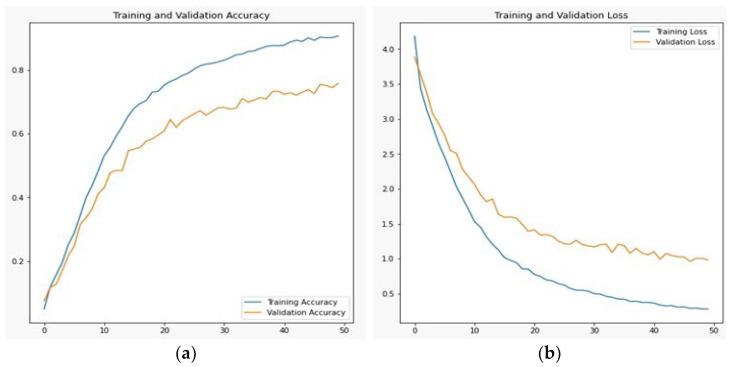
MobileNetV3Large training and validation (**a**) accuracy and (**b**) loss.

**Figure 14 sensors-23-06060-f014:**
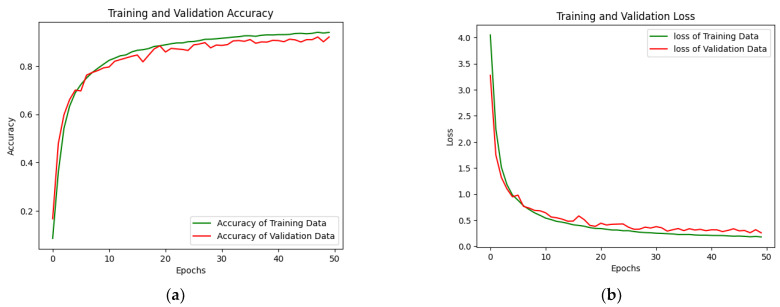
Customized CNN training and validation (**a**) accuracy and (**b**) loss.

**Table 1 sensors-23-06060-t001:** Comparative analysis of the literature review.

Techniques	Pashto Handwritten Character Recognition	Pashto Handwritten Ligatures Recognition	Geometric Variation	Lightweight Classifier	Deep-Learning-Based Classifier	Accuracy in %
Proposed Technique	Yes	Yes	Yes	Yes	Yes	Training = 93.98% Validation = 92.08% Testing = 92.99%
[1]	Yes	No	Unknown	No	Yes	99.6%
[2]	No	No	No	No	No	No accuracy, only dataset
[3]	No	No	No	No	No	Only a printed ligature dataset
[4]	No	No	No	No	No	Not clear
[5]	Yes	No	Yes	Yes	No	93.5% on 730 characters only
[6]	Yes	No	Yes	Yes	No	97.5%
[7]	Yes	No	No	No	Yes	87.6%
[8]	Yes	No	No	No	No	80.34%
[9]	No	No	Yes	No	No	Not Clear
[10]	No	No	No	No	No	Not Clear
[4]	No	No	Yes	No	No	74%
[12]	No	No	No	No	Yes	Not Clear
[13]	No	No	Unknown	No	Yes	9.2%
[14]	Yes	No	No	No	No	Not Clear
[15]	Yes	No	No	No	No	74.8%
[16]	Yes	No	No	No	No	83%
[17]	Yes	No	Unknown	No	No	78%
[18]	No	No	No	No	Yes	Not Clear
[19]	No	No	No	No	No	96.72%
[20]	No	No	Yes	Unknown	Unknown	Not Clear

**Table 2 sensors-23-06060-t002:** Dataset statistics.

Age (Years)	Males	Females	Samples	Designation
12–14	40	20	60	Students
13–14	42	18	60	Students
14–15	25	15	40	Students
28–50	30	10	40	Teachers

Total samples collected: 200.

**Table 3 sensors-23-06060-t003:** Comparative analysis of different deep-learning models.

S: No.	Techniques	Training Loss	Training Accuracy	Validation Loss	Validation Accuracy	Testing Accuracy
1	VGG19	0.1567	0.9467	0.7993	0.8085	0.7921
2	MobileNetV2	0.2327	0.9328	0.6854	0.8024	0.8169
3	MobileNetV3Large	0.2799	0.9062	0.9823	0.7576	0.7999
4	Customized CNN	0.1783	0.9398	0.2573	0.9208	0.9299

## Data Availability

The data presented in this study are available on request from the corresponding author.

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
