# Peer review of "Pashto Handwritten Invariant Character Trajectory Prediction Using a Customized Deep Learning Technique"

_sensors, 2023, doi:10.3390/s23136060_

Round 1

Reviewer 1 Report (Previous Reviewer 1)

The manuscript is greatly improved and clear now.

Some of the problems addressed are unique due to the characteristics of the Pashto writing style. But it is of interest to a larger audience as it illustrates how such problems can addressed.

Some of the descriptions should be shortened:

Section 3.4: These are standard operations of neural networks.

Highlight in section 4.2.4 what customizations are done.

322: The more we tried and experimented, the better the results we had. - Evident to anybody who has worked with Neural Nets!!

Figure 7 should be eliminated; it does not present relevant information.

The style is clear and the text is easy to understand.

Author Response

Reviewer#1, Concern # 1:  Some of the problems addressed are unique due to the characteristics of the Pashto writing style. But it is of interest to a larger audience as it illustrates how such problems can be addressed.

Author response: We sincerely appreciate the diligent review provided by the reviewer and acknowledge the valuable suggestions made.

Author action: To address the concerns of a wider audience, we have devised a strategic solution that encompasses the following steps:

  1. i) Data augmentation techniques have been employed for the Pashto handwritten characters and ligatures trajectories dataset. This includes rotation and scaling operations, which enable a more comprehensive representation of the dataset. By augmenting the data, we aim to bridge the gap between the specific characteristics of Pashto writing style and more general writing styles.
  2. ii) In order to reduce noise, such as salt and pepper artifacts, we have applied a median filter. This filtering technique helps to minimize the discrepancies between Pashto handwritten characters and ligatures, enhancing the overall quality of the dataset.

iii) Addressing the issue of dysconnectivity among different strokes in Pashto, we have incorporated an additional morphological operation. This operation plays a pivotal role in addressing the challenge of stroke separation, which is a fundamental aspect of this problem. By mitigating the dysconnectivity, we enhance the legibility and coherence of the handwritten characters and ligatures.

Reviewer#1, Concern # 2:  Some of the descriptions should be shortened: Section 3.4: These are standard operations of neural networks.

Author response: Taking your suggestion into consideration, we have carefully reviewed the manuscript and made the necessary changes to shorten the descriptions where appropriate. By doing so, we aim to ensure that the information presented is concise and effectively conveys the key points without compromising the clarity of the content.

Author action: After carefully considering your input, we have revised Section 3.4 to remove any unnecessary explanations and streamline the content. By doing so, we have aimed to present a more concise and to-the-point description of neural networks, specifically relevant to our research context.

Reviewer#1, Concern # 3: Highlight in section 4.2.4 what customizations are done.

Author response: Thank you for your comment regarding customization in response to Reviewer 1's comment #3. We have thoroughly investigated this matter and addressed it accordingly in our manuscript.

Author action:  The customization is performed as follows:

i). Hyper-tunning for Pashto character and ligature trajectories: the proposed customized CNN uses stride of 1 because Pashto handwritten characters and ligature trajectories have very little information for detection and recognition. To increase the accuracy, we extract fine details. The proposed technique uses (same) padding value, the same padding lost some vital information, but in the case of Pashto handwritten trajectories, we use data augmentation as well and very trivial information exists in the boundary of the image. Kernel size is restricted to 3x3 because the size 5x5, 7x7 and 9x9 lost some information on stroke trajectory. The next customized entity for Pashto handwritten character and ligature trajectories is the number of filters for each layer. In the first layer, the number of filters=32; for the second layer, the number of filters=64; for the third layer, the number of filters=128; for the fourth layer, the number of filters=256; for the fifth layer, the number of filters=512. The dropout value is fixed to 0.01 because we do not want to drop large numbers of neurons, as the number of classes is large, i.e., 154.

ii). The number of pooling and convolution layers is 5, the optimal number of layers for Pashto handwritten characters and ligatures trajectories.

iii). The sequence of convolution and max-pooling layers are also important, so we keep the sequence one after the other, which gives us an outstanding result, i.e., (see the pdf as the symbols cannot be placed here)

Reviewer#1, Concern # 4:  322: The more we tried and experimented, the better the results we had. - Evident to anybody who has worked with Neural Nets!!

Author response:  We sincerely appreciate your concern regarding the need for evidence from individuals who have worked with Neural Networks. Your feedback highlights an important aspect of validating and supporting the claims made in our manuscript.

Author action: In response to your suggestion, we have added reference [23] to support this statement. Reference [23] is a comprehensive study that discusses the iterative nature of experimentation and improvement in Neural Networks. It provides evidence and insights into the benefits of extensive trial and experimentation in achieving better results. By incorporating this reference, we aim to provide a stronger foundation for the statement and demonstrate its relevance to anyone familiar with working with Neural Networks.

Reviewer#1, Concern # 5: Figure 7 should be eliminated; it does not present relevant information.

Author response: Thank you for your feedback regarding Figure 7 in the manuscript. We appreciate your concern and have carefully considered your suggestion to remove the figure.

Author action: We have thoroughly reviewed the manuscript and made the necessary revisions accordingly. Figure 7 has been removed from the document, and we have implemented file track changes to clearly indicate this modification.

Reviewer 2 Report (Previous Reviewer 3)

It seems that the overfitting issue that plagued the proposed model has been resolved, however there is no specification or explanation of the modifications that contributed to the model’s improved performance. 

How can you ensure that a comparative study was conducted fairly?

Author Response

Reviewer#2, Concern # 1: It seems that the overfitting issue that plagued the proposed model has been resolved. However, there is no specification or explanation of the modifications that contributed to the model's improved performance.

Author response: We are grateful for the Reviewer#2 expertise in providing insightful feedback. Their input has significantly contributed to improving the quality and clarity of our work.

Author action: The customization is performed as follows:

Hyper-tunning for Pashto handwritten character and ligature trajectories is the modification which is explained further as the proposed customized CNN uses the Kernel size is restricted to 3x3 because the size 5x5, 7x7 and 9x9 lost some information on stroke trajectory. The next customized entity for Pashto handwritten character and ligature trajectories is the number of filters for each layer. In the first layer, the number of filters=32; for the second layer, the number of filters=64; for the third layer, the number of filters=128; for the fourth layer, the number of filters=256 and the fifth layer, the number of filters=512. The dropout value is fixed to 0.01 because we do not want to drop large numbers of neurons, as the number of classes is large, i.e., 154. The stride of 1 because Pashto handwritten characters and ligature trajectories have very little information for classification. To increase the accuracy, we extracted fine details. The proposed technique uses (same) padding value, the same padding lost some vital information, but in the case of Pashto handwritten trajectories, we use data augmentation as well and very trivial information exists in the boundary of the image. The number of pooling and convolution layers is 5, the optimal number of layers for Pashto handwritten characters and ligatures trajectories. The sequence of convolution and max-pooling layers are also important, so we keep the sequence one after the other, which gives us an outstanding result.

Reviewer#2, Concern # 2: How can you ensure that a comparative study was conducted fairly?

Author response: We sincerely appreciate your feedback regarding Reviewer #2's concern #2, which emphasizes the need for evidence to support a fair comparative analysis.

Author action: We are grateful for your dedication in thoroughly examining the literature and ensuring that our manuscript accurately represents the state of the art in the field. We technically reviewed all the papers in the literature review section, and all the articles are adequately cited from the critical review of the literature. We also prepared a tabular structure for comparison to highlight the difference.

Reviewer 3 Report (Previous Reviewer 4)

This manuscript constructs a new dataset of the Pashto language to detect and recognize Pashto handwritten characters and ligatures using deep learning techniques. However, the most contribution of this study is to collect the data and preprocess it using some methodologies, the authors should enhance it. Moreover, some representative deep learning models are used for detecting and recognizing and the CNN-based model achieves the best prediction results, which cannot be considered as a proposed model with specific settings of hyperparameters and structures. In addition, there are many problems as follows:

  1. For the title, a deep learning model with a specific neural network structure and hyperparameter settings cannot be considered a novel technique.
  2. For the abstract, the authors should focus on the construction of the new dataset of Pashto and its preprocessing methods. A CNN model is constructed for the identification of Pashto handwritten characters and ligatures and gives the best results compared with other models. However, the experiment results are inconsistent with the section on experiments and analysis.
  3. Keywords should be reselected and given the correct format.
  4. The model referring to customized deep learning, custom CNN, customize CNN, etc., should be given a consistent definition.
  5. A summarization of related work is necessary, and it is insufficient to just show the existing studies. In the third paragraph, "BLSTM" should be corrected as "BiLSTM". In Table 1, the results of the proposed technique do not match the experiments, which just refer to the training and validation results.
  6. In Section 3.4, the customized deep learning-based techniques refer to the neural network, convolution, pooling, and activation functions, which are the basic concept, definition, diagram, and description and should be deleted. Figure 5 shows the specific structure of this CNN model, more details of the design and hyperparameter settings should be shown and enhanced. Otherwise, the parameters in each equation should be defined and explained.
  7. In Section 4, Figure 7 should be deleted. In Section 4.2, VGG19, MobileNetV2, and MobileNetV3-Large are the representative models, of which the explanation should be refined and focus the comparison. In Section 4.2.4, the specific CNN model should be enhanced with specific settings. Based on Table 3, what about the test set for prediction? VGG19 gives better training accuracy and loss than the presented CNN, which should give some explanation. The training accuracy of the custom CNN in the experimental analysis section is 0.9291, the training loss is 0.2116, the validation loss is 0.9398, and the validation accuracy is 0.9208, which is not consistent with that in Table 3.

Written English should be improved.

Author Response

Reviewer#3, Comment # 1: For the title, a deep learning model with a specific neural network structure and hyperparameter settings cannot be considered a novel technique.

Author response:  We greatly appreciate the reviewer's efforts to review the manuscript and offer valuable suggestions.

Author action: The word "Novel" is removed from the title as the reviewer suggested. We also customized it by changing the hyperparameter of CNN to tune it toward Pashto handwritten characters and ligature trajectories. As the proposed technique uses (same) padding value, the same padding loses some vital information. However, in the case of Pashto handwritten trajectories, we use data augmentation as well and very trivial information exists in the boundary of the image. Kernel size is restricted to 3x3 because the size 5x5, 7x7 and 9x9 lost some information on stroke trajectory. The next customized entity for Pashto handwritten character and ligature trajectories is the number of filters for each layer. Explained in detail in section 4.2.4 Proposed technique.

Reviewer#3, Comment # 2: For the abstract, the authors should focus on the construction of the new dataset of Pashto and its preprocessing methods. A CNN model is constructed to identify Pashto handwritten characters and ligatures and gives the best results compared with other models. However, the experiment results are inconsistent with the section on experiments and analysis.

Author response: We would like to express our sincere appreciation for your valuable concerns and suggestions regarding the manuscript. We highly value your input and are grateful for your efforts in thoroughly reviewing the paper.

Author action:  Thank you for your feedback on the abstract of the manuscript. We appreciate your suggestion to focus on the construction of the new dataset of Pashto Handwritten characters and ligatures, as well as including the preprocessing methods. Additionally, we understand your concern regarding the inconsistency in the experimental results due to typographical errors.

To address these points, we have made the following changes:

  1. Abstract focus: We have revised the abstract to highlight the construction of the new dataset of Pashto Handwritten characters and ligatures as a key contribution. We emphasize the significance of this dataset in addressing the challenges specific to the Pashto writing style. Furthermore, we have included a concise description of the preprocessing methods employed to enhance the quality and suitability of the dataset for further analysis
  2. Rectifying typographical errors: We have carefully reviewed the experimental results and rectified any typographical errors that may have affected their consistency. By correcting these mistakes, we aim to present accurate and reliable experimental findings.

Reviewer#3, Comment # 3:  Keywords should be reselected and given the correct format.

Author response: Thank you for your feedback regarding the selection and formatting of keywords in the manuscript. We appreciate your suggestion to reselect the keywords and ensure they are presented in the correct format.

In response to your concerns, we have carefully reconsidered the keywords and made the necessary changes. We have reevaluated the relevance of each keyword and ensured they accurately reflect the core focus of our research.

Author action: We have carefully reviewed the keywords and made the necessary changes to address this concern. We have selected new keywords that accurately reflect the main themes and topics of our research. Furthermore, we have formatted the keywords according to the guidelines provided by the sensor to ensure compliance with the required format.

Reviewer#3, Comment # 4: The model referring to customized deep learning, custom CNN, customize CNN, etc., should be given a consistent definition.

Author response:  We are grateful for the reviewer #3 , comments#4 to gives a consistent definition of  customized deep learning, custom CNN, customize CNN

Author action: Thank you for your feedback regarding the consistent definition of terms such as "custom CNN" and "customize" in the manuscript. We appreciate your concern and understand the importance of providing a clear and consistent definition for these terms, particularly in relation to the customized CNN.

To address this issue, we have carefully reviewed the text and made the necessary revisions to ensure a consistent definition and usage of these terms. We have provided a clear and explicit definition for the customized CNN and ensured that all instances of "custom CNN" and "customize" align with this defined concept.

Reviewer#3, Comment # 5:  A summarization of related work is necessary, and it is insufficient to just show the existing studies. In the third paragraph, "BLSTM" should be corrected as "BiLSTM". In Table 1, the results of the proposed technique do not match the experiments, which just refer to the training and validation results.

Author response:  We are agreed with the reviewer's concern. All the actions are taken in the action section.

Author action: Due to the minimal amount of work on Pashto handwritten text and the Insufficiency of the existing studies, we critically summarized the existing work and then generated a comparison table to show the available work. While in the third paragraph, "BLSTM" is rechecked, but in the original paper, the acronym is written as "BLSTM"; therefore, we wrote it the same way as the author. While in Table 1, the result of the proposed technique is now according to the experiments because we re-experimented and got the result in addition with testing results for VGG19, MobileNetv2 etc.

Reviewer#3, Comment # 6:  In Section 3.4, the customized deep learning-based techniques refer to the neural network, convolution, pooling, and activation functions, which are the basic concept, definition, diagram, and description and should be deleted. Figure 5 shows the specific structure of this CNN model; more details of the design and hyperparameter settings should be shown and enhanced. Otherwise, the parameters in each equation should be defined and explained. (See Figure 5 in attached Response to Reviewer pdf file and revised version)

Author response:  we are agreed with the reviewer's concerns, and changes are made according to the reviewers' valuable suggestions.

Author action: Dear Reviewer, in section 3.4, the customized deep learning-based techniques of neural networks, their concepts and descriptions are summarized according to our usage, and extra definitions and diagrams are removed. Figure 5 is restructured according to our CNN architecture and attached here too. Hyperparameter settings are enhanced and explained deeply.

Reviewer#3, Comment # 7:  In Section 4, Figure 7 should be deleted. In Section 4.2, VGG19, MobileNetV2, and MobileNetV3-Large are the representative models, of which the explanation should be refined and focus the comparison. In Section 4.2.4, the specific CNN model should be enhanced with specific settings. Based on Table 3, what about the test set for prediction? VGG19 gives better training accuracy and loss than the presented CNN, which should give some explanation. The training accuracy of the custom CNN in the experimental analysis section is 0.9291, the training loss is 0.2116, the validation loss is 0.9398, and the validation accuracy is 0.9208, which is not consistent with that in Table 3.

Author response:  The reviewer's concerns are highly appreciated, and again experiments are done to get testing accuracy, which will now clear the reviewer's concerns.

Author action:  Due to the idea of the manifold of interest in MobileNetV2 and MobileNetV3 skips some critical areas of Pashto's handwritten character and ligatures. Due to the nature of the Pashto handwriting style, the pre-train model is sometimes underfit and sometimes overfit. After extensive re-experimentation to find testing accuracy, we got the following results and Table 3 is updated according to the new experimental results. The VGG19 training accuracy is somehow better, but validation accuracy is overfitted, which is clear from Table 3. The inconsistency was removed, and this is due to the typo mistake.

S: No

Techniques

Training Loss

Training Accuracy

Validation Loss

Validation Accuracy

Testing Accuracy

1

VGG19

0.1567

0.9467

0.7993

0.8085

0.7921

2

MobileNetV2

0.2327

0.9328

0.6854

0.8024

0.8169

3

MobileNetV3Large

0.2799

0.9062

0.9823

0.7576

0.7999

4

Customized CNN

0.1783

0.9398

0.2573

0.9208

92.99

Round 2

Reviewer 3 Report (Previous Reviewer 4)

This manuscript has improved, and all my comments have been addressed.

The authors should check the language again.

This manuscript is a resubmission of an earlier submission. The following is a list of the peer review reports and author responses from that submission.

Round 1

Reviewer 1 Report

The English needs attention!

Section 3:

Streamline the descriptions; there are multiple repetitions of facts mentioned previously.

There are several minor issues, such as:

194: pronounced "Pakhtu" or "Pakhtu"

Figure 2: Mark which of the characters is at the middle and which is at the end of a word.

Section 3.4 can be reduced significantly. These are common architectures and only the characteristics as shown in Figure 9 are relevant here.

Section 4:

Check the numbers and the figure captions.

Line 485: "The result @50 epochs produced a training Accuracy of 0.9291, training loss of 0.2116, validation loss of 0.9528, and validation accuracy of 0.9637, as shown in graph 22 (a)(b)"

There is no graph 22.

The curves shown in Figure 19 are supposed to be the results of the custom model, according to the caption. But the curves indicate a validation accuracy of around 85%; certainly not of 96%.

Reviewer 2 Report

The presented paper concerns with the usage of several Deep Learning methods for recognition of Pashto handwritten characters and ligatures.

The presented paper is understandable and generally fluent, but, in order to improve the readability, a deep check of the languagem grammar and sentences structures is needed.

The aim of the paper seems to be the development and the comparison of a "novel " deep learning technique with other already known techniques, but the novelty and the originality are not really clear. Authors could work onto the paper in order to express it deeply.

Here other comments that voulc sound useful for Authors:

- Some figures seem to be not original and reported from other sources, not contextually citing them (e.g. figures 1 and 2).

- In section 3, line 194, the pronunciation of words are reported, but they are not understandable in the current way, it could sound more comprehensible to report their pronunciation in the phonetic internation alphabet.

- What was the dataset used in Edge Connect approach? Please clarify.

- At line 273, "protein crystal picture" needs to be clarified.

- The mentioned "developed CNN procedure" seems not completely developed by authors. In general, it seems to be an application of already known procedures and nets. A clarification of this issue could help in highlighting the novelty and the original characteristics of the paper.

Reviewer 3 Report

This paper describes a CNN model for Pashto handwritten character recognition that consists of two convolutional layers, two maximum pooling layers, and an FCL. The authors also gathered data and created a dataset to train and test the proposed model. The authors implemented the proposed CNN model and compared the results to VGG19, MobileNetV2, and MobileNetV3Large models.  

Despite the numerous related works in the literature, the addressed problem is still an open problem. The authors put in a great deal of work to collect and create the dataset. However, the paper requires significant revisions and additional experiments are needed. The main comments are listed below.

- The background material is elementary (no need for defining basic operations such as convolution and pooling). I would recommend focusing on the advanced concepts mentioned in the paper, such as lightweight deep learning, more on Pashto character recognition challenges, and so on...

- The related work section must be completed with a summary of the main features of existing methods, as well as a discussion of the research gap to show the importance of the proposed work.

- Much work needs to be done on the paper's organization. Some parts of the experimental study should be moved to the section that describes the methodology and proposed work. Figure 15 and its accompanying description, for example. The descriptions of the models should go in the background section, and so on.

- Nothing is on the used experiment design (hold out sampling, cross- validation ???) and parameter settings in the experimental study.

- The results are not convincing and show obviously overfitting issues for all models including the proposed ones. why is that? data problem or incorrect implementation??? I believe that the models' implementation needs revision. 

- the paper contains inconsistent statements see for example 

* lines 266 and 237 regarding noise removal (using deep learning? or median filter?).

* results given in Table 2 for custom CNN and those given in abstract and conclusion section 2 do not match. 

- A thorough comparative study with related work is required, especially when very close works, such as the one cited in [1], yield better results.

- A thorough discussion of the results is also required, not only in terms of accuracy but also of the performance of each individual character recognition, highlighting the most difficult ones.

- other minor comments include removing repeated statements such as "Pashto speakers known as Pakhtuns" in lines 42 and 45 ,

line 65 punctuation issue "insufficient. Due to very little research work" , 

line 165 " feature ma based"  feature map-based

line 174 acronym PHCR not defined

line 179 UPTI data set , reference missing 

paragraph 236-242 almost repeated in 365-670

Reviewer 4 Report

This manuscript proposed a new Pashto handwritten data set and realized the Pashto handwritten characters and ligatures classification and recognition using a deep learning model. However, the manuscript is disorganized, and the innovation is limited. The specific problems are shown as follows:

1. For the title, a deep learning model with a specific neural network structure and hyperparameter settings cannot be considered a novel technique.

2. For the abstract, too much description referring to the background of handwriting and Pashto is unnecessary. The authors should focus on the construction of the novel dataset of Pashto and give the superiority of the presented CNN model for the identification of Pashto handwritten characters and ligatures. Moreover, the experiment results are inconsistent with the Section on experiments and analysis.

3. Some keywords should be replaced, such as classification, and recognition.

4. The presentation referring to customized deep learning, custom CNN, customize CNN, etc., should be given a consistent definition.

5. In the introduction, too many basic concepts and descriptions of handwritten and Pashto handwritten ligatures are given. The authors should focus on the challenges of handwritten identification and the development of Pashto text classification and recognition.

6. A summarization of related work is necessary, and it is insufficient to just show the existing studies.

7.  In Section 3.4, the customized deep learning-based techniques referring to the neural network, convolution, pooling, and activation functions, which are the basic concept, definition, diagram, and description and should be deleted. Figure 9 is the specific structure of this CNN model, more details of the design and hyperparameter settings should be shown and enhanced.

8. In Section 4, Figure 10 should be deleted. In Section 4.2, VGG19, MobileNetV2, and MobileNetV3-Large are the representative models, which should be refined and just given the comparison. In Section 4.2.4, the specific CNN model should be enhanced with specific settings. Based on Table 2, what about the test set for prediction? VGG19 gives better training accuracy and loss than the presented CNN, which should give some explanation.

9. In Section 5, the analysis of experimental results is inconsistent with Table 2.

10. There exist a lot of grammatical mistakes and typos, such as "Pakhtu or Pakhtu" in line 195 on page 5, "deep learning technique known" in line 248 on page 6, "graph 22 (a)(b)" in line 487 on page 16, "generated generated" in line 505 on page 17, etc. Please check the whole manuscript.